Report

# Meta-analysis of clinical metabolic profiling studies in cancer: challenges and opportunities

Jermaine Goveia[1,2], Andreas Pircher[1,2], Lena-Christin Conradi[1,2], Joanna Kalucka[1,2], Vincenzo Lagani[3], Mieke Dewerchin[1,2], Guy Eelen[1,2], Ralph J DeBerardinis[4], Ian D Wilson[5] & Peter Carmeliet[1,2,*]

## Abstract

Cancer cell metabolism has received increasing attention. Despite a boost in the application of clinical metabolic profiling (CMP) in cancer patients, a meta-analysis has not been performed. The primary goal of this study was to assess whether public accessibility of metabolomics data and identification and reporting of metabolites were sufficient to assess which metabolites were consistently altered in cancer patients. We therefore retrospectively curated data from CMP studies in cancer patients published during 5 recent years and used an established vote-counting method to perform a semiquantitative meta-analysis of metabolites in tumor tissue and blood. This analysis confirmed well-known increases in glycolytic metabolites, but also unveiled unprecedented changes in other metabolites such as ketone bodies and amino acids (histidine, tryptophan). However, this study also highlighted that insufficient public accessibility of metabolomics data, and inadequate metabolite identification and reporting hamper the discovery potential of meta-analyses of CMP studies, calling for improved standardization of metabolomics studies.

**Keywords** cancer; meta-analysis; metabolic profiling; metabolomics
**Subject Categories** Cancer; Metabolism; Systems Medicine

## Introduction

Clinical metabolomics investigates how metabolite levels are altered in various (patho)physiological conditions, often with the objective to find new roles of metabolism in disease, to discover novel metabolic drug targets, or to identify biomarkers (Fernie *et al*, 2004). Hopes have been raised that clinical metabolic profiling (CMP) could reshape our understanding of cell biology and pathophysiology, and even improve clinical practice (Patti *et al*, 2012). However,

apart from a few high-profile discoveries (Dang *et al*, 2009; Wang *et al*, 2011), these expectations have not been fully met and the impact of CMP studies has remained relatively modest (Sevin *et al*, 2015). This has raised concerns about the robustness, consistency, and translational potential of CMP studies (Gika *et al*, 2014). In contrast, the clinical impact of transcriptomics, genomics, and proteomics has greatly benefited from standardized data reporting and accessibility, permitting efficient data mining and quantitative meta-analyses (Fernie *et al*, 2004; Rosenberg *et al*, 2010; Hu *et al*, 2013a,b; Nilsson *et al*, 2014).

Tools have been developed to deposit CMP results in databases for managing (meta)data of metabolome analyses, but not for performing meta-analyses (Haug *et al*, 2013; Ara *et al*, 2015; Salek *et al*, 2015; Rocca-Serra *et al*, 2016). Surprisingly, however, even though descriptive meta-studies that overview CMP data have been reported (Shah *et al*, 2012; Abbassi-Ghadi *et al*, 2013; Huynh *et al*, 2014; Nickler *et al*, 2015; Guasch-Ferre *et al*, 2016), not a single study performed a quantitative meta-analysis, in particular in cancer. Nonetheless, the aggregation of information from multiple studies in a meta-analysis leads in many cases to higher statistical (discovery) power and therefore higher impact of individual studies (Green, 2005). It remains undetermined whether a meta-analysis of cancer CMP studies would offer novel insight, since cancer is a heterogeneous disease, and CMP studies greatly vary in (i) how and how many metabolites are measured, identified, and reported; (ii) how such studies are designed; and (iii) whether and how they are validated (Dunn *et al*, 2012). Only very recently, the first in class meta-analysis of CMP was reported. However, this meta-analysis was performed only on a subset of prospective CMP studies in diabetic patients and even though this study associated elevated plasma levels of branched-chain amino acids with the risk of developing type 2 diabetes (T2DM), it did not attempt to aggregate and analyze the data of all the metabolites reported in all individual studies (Guasch-Ferre *et al*, 2016).

For genomics, transcriptomics and proteomics data, the availability of raw data such as abundances of transcript and protein levels offers the possibility to compare the datasets in their original form (Brazma *et al*, 2003; Jones *et al*, 2006; Barrett *et al*, 2013). When

---

1 Laboratory of Angiogenesis and Vascular Metabolism, Department of Oncology, KU Leuven, Leuven, Belgium
2 Laboratory of Angiogenesis and Vascular Metabolism, Vesalius Research Center, VIB, Leuven, Belgium
3 Computer Science Department, University of Crete, Heraklion, Greece
4 Children's Medical Center Research Institute, University of Texas Southwestern Medical Center, Dallas, TX, USA
5 Department of Surgery and Cancer, Imperial College, London, UK
*Corresponding author. Tel: +32 16 37 32 02; Fax: +32 16 37 25 85; E-mail: peter.carmeliet@vib-kuleuven.be

---

 

such quantitative data are not available, the results can still be analyzed in a semiquantitative meta-analysis by vote counting, a technique that is generally applicable and does not rely on the availability of raw data (Rikke *et al*, 2015). Vote counting has been successfully used in previous meta-analyses to identify metabolic targets, the expression of which was consistently deregulated across multiple cancer types (Nilsson *et al*, 2014).

In this study, focusing on cancer, we retrospectively generated a curated list of metabolites, based on MEDLINE search filter criteria, from previous CMP studies in cancer patients published during 5 recent years, and used vote counting to perform a semiquantitative meta-analysis. The primary goal of this study was to assess whether public accessibility of metabolomics data, metabolite identification and reporting were sufficient to obtain, novel insight in consistent metabolite changes in cancer patients. It was not the primary goal of this study to identify new metabolic drug targets or biomarkers, or to create a comprehensive, widely useful cancer metabolite database *per se*. Rather, we explored whether a meta-analysis of CMP studies is feasible, and how these CMP studies can be improved to meet the same standards as routinely used in the genomics, transcriptomics and proteomics fields.

## Results

### Compilation of a curated cancer metabolomics dataset

Since deposition of metabolomics data in publicly available repositories is generally not required by scientific journals to date, comprehensive datasets for meta-analysis have to be created by alternative approaches, for instance, by retrospective manual curation. We therefore conducted a systematic review of the literature to identify all CMP studies in cancer published between June 2010 and June 2015. For all studies, we extracted data on key methodological parameters using a pre-defined data extraction protocol such as the type of disease, number of patients included, the analytical platform, outcome measures, the level of metabolite identification, and major findings among others. We also extracted information on all reported metabolites, such as raw abundance, fold change, and whether a metabolite was up- or downregulated in cancer. Because the vast majority of studies reported metabolites using ambiguous common names but not unique identifiers, we used (bio)informatics tools to extensively curate the extracted data of each study (see Materials and Methods). The resulting collection contains curated quality-checked data of 136 cohorts reported in 126 studies, spanning 18 tumor types and over 5,300 "disease versus control" comparisons of approximately 1,900 unique metabolites in blood, urine, and tumor tissue (denoted as "tissue" from here onwards) from an estimated 21,000 individuals (Fig 1 for study outline; Table EV1; see Materials and Methods for details).

### Clinical metabolic profiling: methods and limitations

#### Data reporting
To assess how complete the reporting of the measured metabolites was done relative to all previously reported metabolites, we indicated for each study whether the metabolite was reported or not. Current metabolic profiling technologies are capable of measuring tens to hundreds of metabolites. However, surprisingly, most individual studies published only a very small subset of all earlier reported metabolites. This is clearly visible from the heat maps shown in Fig 2A (for blood) and Fig EV1A and B (for tumor tissue and urine), where a dark blue mark denotes that the metabolite was reported to be increased or reduced in cancer. From the abundant white "empty" space, it is obvious that reporting of metabolites was highly incomplete. Even metabolites associated with a major chemical class (such as amino acids, carbohydrates) were reported on average in only 6.4% of the studies. This finding can be explained in part by the use of different profiling methodologies across studies.

Notably, however, the majority of studies reported only metabolites presumably deemed to be relevant to the authors and used heterogeneous statistical outcome measures without providing full datasets (Table 1). In fact, even though CMP studies have the potential to assess many hundreds of metabolites, the median number of reported metabolites per study was only 22 (Table 1). Moreover, while most, but not all, studies provided information regarding the magnitude of the change ("effect size"), only 22.8% of the studies reported measures of variance (Table 1). Also, only a mere 18.7% of all studies reported data on all measured metabolites.

Taken together, it appears that in general, metabolite reporting was highly incomplete and presumably subjective. This is in sharp contrast to the genomics, proteomics, and transcriptomics analyses, where full dataset deposition in publically available repositories is often required.

#### Metadata reporting
Cancer is a heterogeneous disease. Therefore, cancer patients are often clinically stratified based on demographic factors (such as age and gender), tumor staging, histological parameters, molecular tumor characteristics, treatment response, and others. However, it is generally acknowledged that clinical and experimental metadata reporting is problematic in most CMP studies, thus not only for cancer (Ara *et al*, 2015). Indeed, we noticed that metadata have been only scarcely reported, even though specialized databases exist to collect such data from metabolic profiling experiments specifically (Ara *et al*, 2015). This precluded us from factoring patient and tumor heterogeneity into our meta-analysis.

#### Study design
A total of 122 of the CMP studies (96.8%) included in our study employed an observational, cross-sectional research design, in which cancer patients were compared to a control group at a particular time point. While such CMP studies may discriminate between cancer and control and could provide novel insight in disease pathogenesis, such experimental design is not (necessarily) optimal to discover novel metabolic biomarkers for patient stratification. A major goal of modern medicine is to stratify patients for personalized treatment and to identify biomarkers that can predict disease course or treatment response. However, the majority of CMP studies did not consider any factors that could aid in patient stratification other than the presence of disease. Also, biomarker discovery and validation requires a prospective research design in which patients are followed up over time to associate metabolite levels with the course of disease

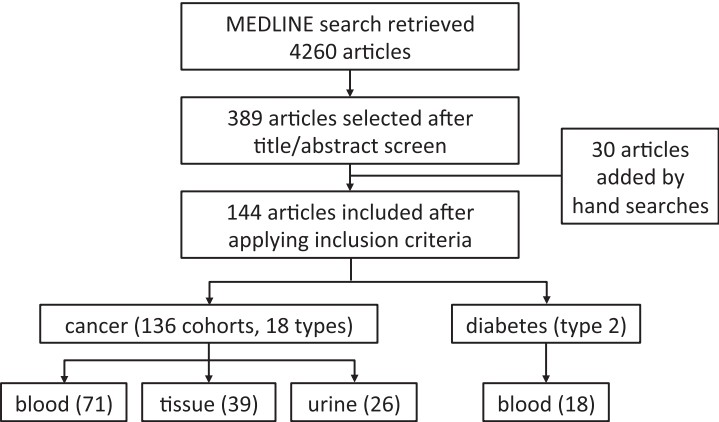

**Figure 1.  Overview of the study design and dataset compilation.**
We included 126 studies in our curated cancer metabolomics dataset, studies that described profiling in multiple cancer types were counted once for each type, giving rise to 136 "cancer versus control" cohorts spanning 18 different cancer types, 71 assessing blood, 39 tissue, and 26 urine. We also included 18 diabetes mellitus type 2 studies to determine whether our vote-counting method could identify distinct metabolic signatures in cancer versus diabetes.

or treatment response. Of the cancer studies we considered, only 4 (3.2%) had a longitudinal research design or compared early with late-stage cancer to assess metabolic alterations over the course of disease.

*Metabolite identification*

Metabolite identification is a major bottleneck in CMP, but is nonetheless essential for adequate biological interpretation of the results. The Metabolomics Standards Initiative (see http://cosmos -fp7.eu/msi for more information) defined four levels of identification, of which only "level one" results in unambiguous annotation (Salek *et al*, 2013). Notably, only half (52%) of the CMP studies provided "level one" metabolite identification for at least a subset of the reported metabolites, and even fewer studies identified all reported metabolites unambiguously, often studies that profiled a small set or specific class of metabolites.

*Clinical or orthogonal validation*

Metabolic profiling produces high-dimensional data (a typical dataset may contain values for hundreds to thousands of metabolites for each sample analyzed). Statistical analysis of such data is prone to type I errors ("false positives"). Therefore, results should be best validated in independent cohorts or verified by using orthogonal models or by using different independent technologies (e.g. transcriptomics, proteomics) and/or targeted analysis. However, only 17.9% of the CMP studies reported validation cohorts and only ~10% used orthogonal models. Even though metabolic profiling is ideally suited to combine with other (orthogonal) omics data, only 6.8% of the studies performed multi-omics analysis (Table EV2). Thus, the findings of the majority of CMP studies remain unconfirmed. In principle, a meta-analysis is useful to validate the results of individual studies in independent cohorts.

**Identification of metabolic signatures in cancer**

Incomplete and heterogeneous reporting of metabolite data and summary statistics prevented us from performing a quantitative meta-analysis and precluded us from determining the average fold changes of metabolite levels across all studies for any metabolite. Also, scarce availability of metadata prevented us from stratifying cancer patients and from assessing an association between metabolite changes and patient or tumor characteristics. These omissions in data reporting likely explain why previous metabolomics meta-studies did not perform statistical aggregation of the results from individual studies (Rocca-Serra *et al*, 2016). However, all studies in our dataset reported the directionality (increased or decreased levels) of the deregulated metabolites. We therefore performed a meta-analysis by vote counting (Rikke *et al*, 2015), a semiquantitative technique that only requires such information, allowing us to include all studies in the analysis for improved statistical power. Nonetheless, our meta-analysis was still (relatively) underpowered, and we obtained only statistical significance for a subset of metabolites, even though other metabolites showed clear trends that could become statistically significant with more power, and hence may be of clinical relevance.

*Meta-analysis approach*

To explore how consistently metabolites are altered across cancer types, we indicated for each metabolite per study whether it was increased (denoted by "+1") or decreased ("−1") in cancer patients relative to controls. These controls were "healthy" individuals without cancer for analysis of blood and urine, and, for tumor tissue, controls included subjects (i) without cancer, (ii) with premalignant lesions, or (iii) with cancer but using adjacent healthy tissue as control. The vote-counting statistic (VCS, reported as VCS/number of reporting studies; Benjamini–Hochberg adjusted *P*-value) assumes a high positive value if the metabolite was consistently increased and conversely a negative value for consistently decreased metabolites. In this context, a zero value implies that the studies provide conflicting evidence on whether the metabolite was decreased or increased. The Benjamini-Hochberg adjusted *P*-value was only calculated for metabolites reported in at least six cohorts. While the statistical power of urine meta-analysis was limited due

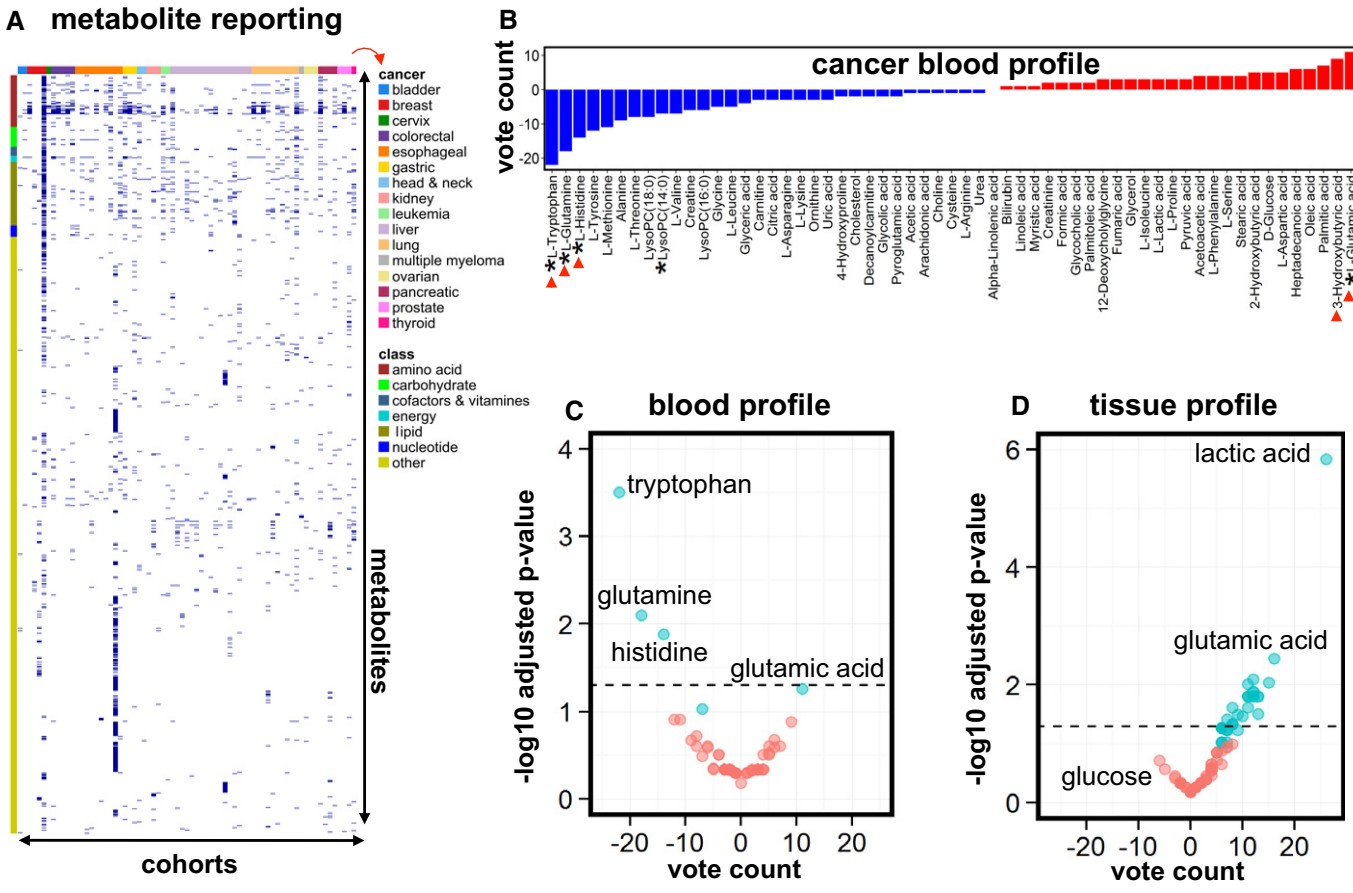

**Figure 2. Metabolite reporting & vote-counting analysis in tissue and blood.**

A    Heat map of all reported cancer blood metabolites (each dark blue mark denotes that the metabolite was reported to be increased or decreased) in various cancer types, illustrating that most individual studies report only a small subset of all previously measured metabolites. It is typically not described whether the metabolites that were not reported (white "empty" space) were not measured, or measured but not reported. Labeling: x-axis (top): ribbon color code, denoting the cancer type (right; indicated by red arrow); x-axis (bottom): cohorts, arranged from 1 to 71; y-axis (right): all 1,206 metabolites reported in at least one of the studies; y-axis (left): ribbon color code, denoting the metabolite class (amino acids, carbohydrates, etc.; "other" refers to all other metabolites than the listed classes). See also Table EV4.

B    Vote counting of cancer blood metabolites (reported in at least six cohorts) showed consistently deregulated metabolites. Blue bars: decreased metabolites; red bars: increased metabolites. An asterisk (*) in front of the name of the metabolite indicates at least a statistical trend (P < 0.1) obtained using the sign test; red arrowheads denote metabolites mentioned in the main text.

C, D    Volcano plots of blood metabolites (C) and tissue metabolites (D), reported in at least six cancer studies, with the vote-counting score on the x-axis and the −log10 adjusted P-value on the y-axis. Cyan indicates deregulated metabolites that show a trend (P < 0.1; for blood, corresponding to metabolites marked with * in panel B; for tissue, corresponding to metabolites marked with * in Fig 3) or statistical significance (P < 0.05; above black dashed horizontal line); red indicates metabolites with a P-value > 0.1. A subset of metabolites is annotated (see Tables EV4 (blood) and EV5 (tissue) for full annotation, vote-counting statistics were calculated using the sign test).

to a small number of studies (Fig EV2; Tables EV1 and EV3), our analysis revealed profiles of consistently deregulated metabolites in blood (Fig 2B and C; Table EV4) and tumor tissue (Figs 2D and 3; Table EV5) across cancer types.

*Cancer-associated metabolic changes*
We then assessed whether the vote-counting method identified particular metabolites that were consistently up- or downregulated in cancer. In agreement with the known increase of glycolysis in cancer cells (Vander Heiden *et al*, 2009), this analysis showed increased tumor lactic acid levels consistently across all cancer types examined (VCS = 26/26, P-value = $1.5 \times 10^{-6}$) (Figs 2D and 3). Interestingly, glutamic acid ranked second (only after lactic acid) among the most increased metabolites in tumor tissue (VCS = 16/18,

P-value = $3.6 \times 10^{-3}$; Figs 2D and 3) and was the most frequently increased metabolite in blood (VCS = 11/15, P-value = $5.5 \times 10^{-2}$; Fig 2B and C). The glutamic acid precursor glutamine was the second most decreased metabolite in blood (VCS = −18/26, P-value = $8.0 \times 10^{-3}$) and was frequently increased in tumor tissue (VCS = 7/13, P-value = $1.1 \times 10^{-1}$; Figs 2B–D and 3). The findings in the blood may indicate systemic depletion of glutamine and other amino acids (see below) as observed in chronic catabolic states (Souba, 1993). Overall, this analysis confirms that the vote-counting method can identify changes in metabolites, which have been previously implicated in cancer cell metabolism (Lunt & Vander Heiden, 2011).

A novel finding was that the ketone body 3-hydroxybutyric acid was upregulated in the blood of cancer patients (VCS = 9/15,

## Table 1. Metabolomics data reporting.

|  | Cancer | | | All cohorts |
|---|---|---|---|---|
|  | Blood | Urine | Tissue |  |
| Median number of reported metabolites | 19 | 19.5 | 29 | 22 |
| Effect size reported (%) | 98.5 | 92 | 100 | 97.9 |
| Variance measure reported (%) | 30.8 | 20 | 10.3 | 22.8 |
| Full dataset available (%) | 23.1 | 8 | 17.9 | 18.7 |

$P$-value = $1.3 \times 10^{-1}$) (Fig 2B). This ketone body has been reported to stimulate tumor growth and has been associated with cancer cachexia (Tisdale & Beck, 1990; Shukla et al, 2014), though its role remains debated (Bonuccelli et al, 2010; Poff et al, 2014; Shukla et al, 2014). We also identified significant deregulation of less investigated metabolites. In the blood, tryptophan (VCS = $-22/26$, $P$-value = $3.2 \times 10^{-4}$) and histidine (VCS = $-14/18$, $P$-value = $1.3 \times 10^{-2}$) were among the top three most decreased metabolites (Fig 2B and C), while they were increased in tumor tissue (VCS = 8/10, $P$-value = $4.6 \times 10^{-2}$ for both tryptophan and histidine) (Fig 3). Interestingly, histidine has been implicated in tumor-associated inflammation (Yang et al, 2011), while the tryptophan metabolite kynurenine (frequently increased in tumor tissue; VCS = 7/9, $P$-value = $5.9 \times 10^{-2}$; Fig 3) suppresses anti-tumor immune responses (Opitz et al, 2011). In addition, both tryptophan and histidine are potential one-carbon donors to tetrahydrofolate, which contributes to nucleotide metabolism and redox homeostasis, perhaps reflecting the augmented proliferative potential of cancer cells. These results indicate that vote counting can identify metabolites that are often up- or downregulated in cancer patients.

### Sensitivity analysis

We performed a sensitivity analysis to assess whether the cross-cancer results were driven/largely influenced by individual cancer

types. To this end, the vote-counting procedure was separately repeated by excluding in turn each cancer type for studies in urine, blood, and tissue. The top deregulated metabolites were consistently deregulated, regardless of which cancer type was taken out of the analysis, confirming that no individual cancer type dominated the analysis (not shown).

### Type 2 diabetes

To determine whether metabolic alterations could also be detected with the vote-counting method in another disease, we constructed a second dataset of blood metabolites in T2DM patients (18 studies, ~1,200 metabolite measurements in estimated 4,000 patients; Table EV1) and repeated our meta-analysis. Due to the limited number of CMP studies, the study was relatively underpowered. Nevertheless, using our approach, we identified a number of validated T2DM biomarkers, including, as expected, glucose (VCS = 7/7, $P$-value = $7.0 \times 10^{-2}$). However, we also observed elevations of the (branched-chain) amino acids leucine (VCS = 9/11, $P$-value = $7.0 \times 10^{-2}$), valine (VCS = 7/11, $P$-value = $1.1 \times 10^{-1}$), isoleucine (VCS = 5/7, $P$-value = $1.4 \times 10^{-1}$), and phenylalanine (VCS = 7/9, $P$-value = $8.8 \times 10^{-2}$) (Fig EV3, Table EV6). Interestingly, these data are consistent with a recent report based on previously published prospective studies that elevated levels of these amino acids are associated with an increased risk to develop T2DM (Guasch-Ferre et al, 2016), thus validating our approach. Furthermore, the same study also found that glycine blood levels were inversely associated with T2DM risk. Of note, we identified glycine as the most decreased metabolite in our analysis (VCS = $-5/7$, $P$-value = $1.4 \times 10^{-1}$) (Fig EV3, Table EV6). These concordances further validate the potential of the vote-counting method to identify changes in metabolites that can be clinically relevant. Another noteworthy observation is that blood levels of 1,5-anhydrosorbitol were consistently reduced in all three studies that reported this metabolite (Table EV6). This metabolite is a clinical biomarker of diabetes and has been developed in a FDA-approved blood glucose test (Halama et al, 2016). Overall, these results indicate that our

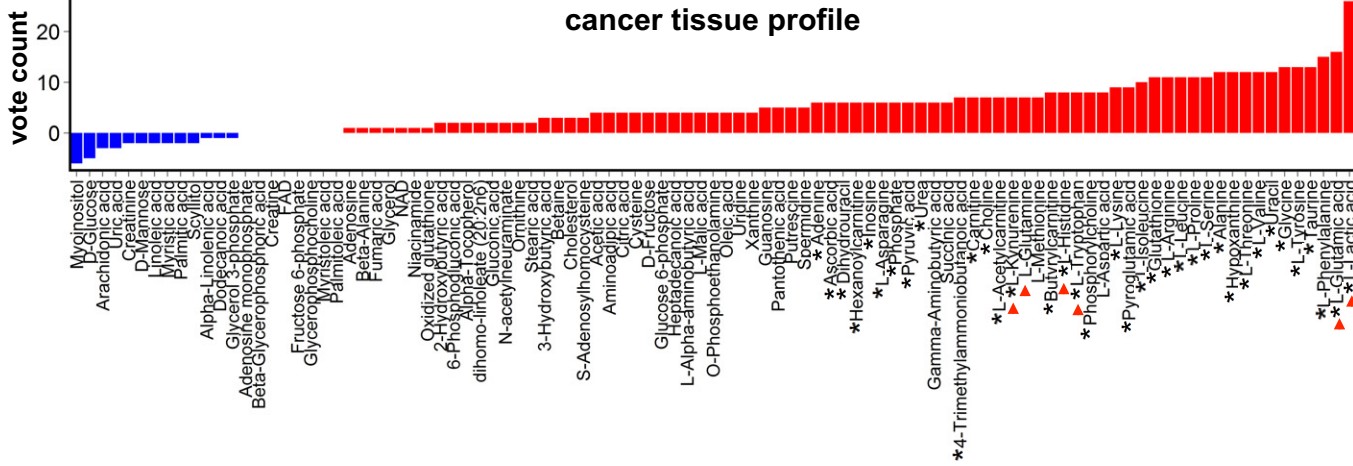

### Figure 3. Vote-counting analysis in cancer tissue.

Vote counting of cancer tissue metabolites (reported in at least six cohorts) showed consistently deregulated metabolites. Blue bars: decreased metabolites; red bars: increased metabolites. An asterisk (*) in front of the name of the metabolite indicates at least a statistical trend ($P < 0.1$) obtained using the sign test; red arrows denote key metabolites mentioned in the main text (see Table EV5 for full annotation).

semiquantitative meta-analysis is capable of identifying distinct metabolite signatures for cancer versus T2DM.

## Discussion

Despite incomplete reporting of CMP data, the semiquantitative meta-analysis approach used in this study was capable of identifying distinct metabolite signatures in cancer and T2DM. Not only did this method provide confirmatory evidence for the up- or downregulation of particular metabolites, previously involved in cancer cell metabolism (Vander Heiden *et al*, 2009; Lunt & Vander Heiden, 2011), but the approach used also identified metabolites that were less well/only minimally appreciated in cancer cell metabolism. While our analysis reveals that meta-analyses of CMP studies provide novel opportunities, our study was also confronted by a number of challenges that need to be addressed in order to maximize the discovery potential of this approach.

### Challenges

Despite the promising results of our analysis, several hurdles limit the amenability of current CMP studies for quantitative meta-analysis: (i) The lack of availability of full datasets reduces the statistical power of data mining approaches and precludes re-analyzing the original data to replicate the results. Moreover, without the availability of full datasets, it is impossible to determine whether a metabolite was measured but not deemed relevant, or not measured, leading to a particularly pernicious bias in outcome reporting that cannot be adequately resolved with current statistical methods. (ii) Scarce reporting of metadata and lack of raw data for each analyzed sample limits the possibilities for stratified analysis, while such analysis of inter-patient heterogeneity is key to modern personalized medicine. (iii) Many studies use cross-sectional or case–control research designs to prove that the metabolic profile in cancer is distinct from controls. Such studies can validate the discriminatory power of the metabolic profiling technology, but the results are not necessarily clinically relevant or applicable to improve personalized medicine. (iv) The use of inconsistent reporting formats renders data extraction highly labor-intensive and error-prone. (v) Ambiguous metabolite annotation and identification limit confidence in the interpretation of data mining outcomes. (vi) The lack of quantitative data (absolute levels of metabolites) precludes the development of reference values to which individual patients can be compared. (vii) The use of a wide variety of profiling technologies and analytical methods complicates direct comparison between studies. And, (viii) the lack of validation in independent replication cohorts, in preclinical models or by orthogonal experiments, is a shortcoming of many profiling studies.

### Opportunities

Nonetheless, despite these limitations, a semiquantitative meta-analysis of a limited number of studies appeared to be sufficient to identify a number of known and new metabolites, some underscored by a significant *P*-value, others more by a trend (likely due to insufficient power). Clearly, standardizing the design, execution, and reporting of CMP studies will only increase the discovery

potential of quantitative meta-analyses. Today, only a minority of CMP data is readily available, and retrospective collection and curation from past publications is an enormous effort. To maximize the value of CMP studies, metabolomics data could be submitted to a public metabolomics repository (Haug *et al*, 2013). Eventually, to improve the feasibility and power of future data mining approaches, the research community and journals can adopt reporting standards that require data annotation in an accessible electronic format (Larsson & Sandberg, 2006; Rocca-Serra *et al*, 2016).

### Limitations

Our study was designed to test whether a meta-analysis could be performed with currently available CMP studies in cancer. Given the aforementioned challenges, the scope of our study was not to discover new metabolic targets *per se*, or to create a database for broad usage. Because comprehensive metabolomics repositories do not exist, we performed a meta-analysis on reported metabolites, an approach that is susceptible to publication bias. In conclusion, the largest limitation is the insufficient number of studies and lack of full dataset availability, which reduced statistical power, prevented detection of metabolic signatures within cancer types, and precluded stratified analyses.

## Materials and Methods

### Rationale and objectives

We performed a meta-analysis of metabolite profiling in cancer with the following objectives: (i) to provide a cross-section of metabolic profiling methods and implementations; and (ii) to assess the potential of metabolomics data mining. The aim of this study was not to identify, nor to functionally verify new metabolic mechanisms, discovered through this meta-analysis.

### Dataset compilation

#### General strategy
Since no datasets of metabolic profiling in cancer exist, we first set out to construct such a dataset. We used a pre-defined search filter to search MEDLINE for studies reporting on metabolic profiling of blood (plasma and/or serum), tissue, and urine in cancer. Next, we included studies based on predetermined in- and exclusion criteria (Table EV7) and extracted metabolite data as well as methodological metadata using a standardized data extraction sheet. In addition, we noted for each metabolite whether it was reported as increased or decreased. The same strategy was used to construct a second dataset of metabolic profiling in type 2 diabetes mellitus (T2DM) in order to assess disease specificity of identified metabolic alterations.

#### Search method for identification of studies
Our first search filter combined the search term "metabolic" and its synonyms with the Boolean operator "OR". Our second search filter combined the term "profiling" and its synonyms with the Boolean operator OR. The third search filter combined the terms "diabetes" and "cancer" with the operator OR. We then combined

these searches with the Boolean operator AND. We added additional filters to specifically exclude studies on metabolic syndrome, review articles, and editorials. We limited the search to studies published from June 1, 2010, to June 1, 2015, in the English literature, to ensure that the data examined reflected current state-of-the-art profiling platforms. This filter retrieved 4,260 references of articles. To identify potentially relevant studies, two reviewers (A.P. and J.G.) independently screened all studies on title and abstract. To further increase coverage and retrieval of relevant studies, we complemented our automated search with manual searches (J.K., L.C.C., and J.G.) in relevant journals and identified 30 additional papers.

### Inclusion and exclusion of studies

Next, three teams of two reviewers assessed the eligibility of the identified papers (A.P./J.G., L.C.C./J.G., J.K./J.G.) by applying the in- and exclusion criteria. Briefly, we considered global metabolic profiling studies performed in humans, in whom metabolic profiling was done on serum, plasma, urine, or tumor tissue. Studies profiling other bodily fluids (such as cerebrospinal fluid and sweat) and feces only or solely in other organisms were excluded. We considered both clinical biomarker studies and studies where metabolic profiling was used as a discovery strategy to identify novel biological mechanisms. We only included studies profiling cancer or T2DM that had a well-defined control group. We further limited inclusion to studies that used either nuclear magnetic resonance spectroscopy (NMR) or mass spectrometry (MS) in a global profiling approach. An overview of included studies is provided in Tables EV8, EV9, EV10 and EV11.

### Quality assessment, data extraction and integrity

For all selected papers, the reviewers used a pre-defined evaluation/data extraction protocol to assess (methodological) study parameters such as the type of disease, number of patients included, the analytical platform, outcome, the level of metabolite identification (see below), and major findings among others. Because the vast majority of studies annotated reported metabolites using ambiguous common names and did not use unique identifiers, which limits bioinformatics possibilities, we used specific algorithms of the OpenRefine software to standardize spelling and the ConvertTool of the MetaboAnalyst 3.0 software to assign unique identifiers if available. Retrospective manual curation of metabolomics data from published studies is time-consuming and relatively error-prone, but currently the only available method to construct a comprehensive metabolomics dataset. We performed data quality checks at each step of the dataset construction and randomly re-reviewed 25% of the included references post-dataset construction.

### Metabolite identification levels as defined by the Metabolomics Standards Initiative

Level "one" (identified metabolites) identification necessitates that two or more orthogonal properties of an authentic chemical standard, analyzed in the researcher's laboratory, are compared to experimental data acquired in the same laboratory with the same analytical methods. By contrast, annotation of level "two" (putatively annotated compounds) and level "three"

**The paper explained**

**Problem**

It has become increasingly clear that cellular metabolic alterations are involved in the pathogenesis of multiple diseases, including cancer. Clinical metabolic profiling (CMP) offers the opportunity to characterize the metabolome and is being increasingly used to identify disease-associated metabolic signatures. Metabolic profiling of body fluids and excretions such as blood, urine, and feces was anticipated to identify novel biomarkers, while profiling of diseased tissue was expected to provide insight into molecular pathogenesis. However, despite technological advances and increasing use of CMP, progress and clinical translation of these data have been more modest than expected. Surprisingly, a meta-analysis of CMP has not been performed, even though (i) such studies could provide insight into current methods and application of metabolic profiling; and (ii) similar meta-analyses have contributed to the impact in the genomics, transcriptomics and proteomics fields. Here, we performed a meta-analysis of CMP in cancer.

**Results**

We manually compiled a dataset of cancer studies published during five recent years. Insufficient raw data and metadata availability and reporting prevented quantitative meta-analysis. We therefore performed a semiquantitative meta-analysis by vote counting (comparing the number of studies that report a metabolite to be increased or decreased in order to identify metabolites that are consistently deregulated across studies) to demonstrate the potential of CMP data mining, as highlighted by the identification of several known but also previously less appreciated metabolic alterations in cancer.

**Impact**

Our results suggest that the clinical impact of metabolic profiling could be improved by adhering to standards for designing clinical studies, more extensive validation of the results and, most importantly, by improved metabolomics data reporting (and metabolite identification) and deposition of full datasets in public repositories for reuse in meta-studies. Scientific journals may facilitate this process by demanding full dataset availability as is already required for genomics, transcriptomics, and proteomics data.

(putatively characterized compound classes) does not require matching to data for authentic chemical standards acquired within the same laboratory. Level "four" refers to unidentified compounds.

## Statistical analyses

### Vote-counting procedure

Because there are no standardized approaches available for metabolomics meta-analysis and full quantitative data (fold changes, abundances, etc.) are lacking, we devised a statistical approach to quantify metabolite deregulation. For each study, a score of "+1" or "−1" was assigned to each reported metabolite, depending on whether the metabolite was found increased or decreased, respectively, regardless of the fold change. Metabolites that were reported more than once per study, for example, as measured by different methods were only annotated once. For these metabolites, we determined directionality (whether it was increased or decreased) by applying vote counting within the study. This resulted in an assigned score of "0" if the study reported it

equally often as increased and decreased, because directionality could not be determined (21, 42, and 1 metabolite in blood, urine, and tissue samples, respectively). We then performed a semiquantitative meta-analysis by vote counting, that is, summing the scores for each metabolite, to quantify deregulation (i.e. the consistency with which a metabolite is reported to be increased or decreased across studies). The vote-counting statistic (VCS, reported as VCS/number of reporting studies; Benjamini–Hochberg adjusted *P*-value) assumes a high positive value if the metabolite is consistently increased, and conversely a negative value for consistently decreased metabolites. In this context, a zero value implies that the studies provide conflicting evidence on whether the metabolite is increased or decreased.

We also used the sign test to assign a *P*-value to assess whether the vote-counting results were merely due to chance. The null hypothesis was that the directionality of deregulation of a given metabolite is not consistent across studies. Under this null hypothesis, we assumed that studies would report the metabolite as either increased or decreased with approximately equal probability. With this assumption, we could model the global score for a given metabolite as the result of a binomial process, where (i) the number of trials equals the number of studies reporting the metabolite, (ii) success is defined as a study with the same sign of the global score, and (iii) the probability of success is set to 0.5. The one-tailed *P*-value assessing the probability of obtaining an equal or greater (in absolute terms) global score under the null hypothesis was obtained from the binomial distribution and adjusted for multiple testing with the Benjamini–Hochberg procedure (Benjamini & Hochberg, 1995). The significance of metabolites reported in 5 or fewer studies was not assessed for lack of statistical evidence.

An alternative, permutation-based test was also employed. For each metabolite, the null distribution was built by computing several times the global score on a simulated set of reporting studies. Each simulated set was built by randomly assigning with probability 0.5 an up/down directionality to each study reporting the metabolite. A one-tailed *P*-value was computed by comparing the original global score against its respective null distribution. The results of this permutation test were almost identical to the *P*-values provided by the binomial test (correlation > 0.9) and are thus not reported.

**Expanded View** for this article is available online.

## Acknowledgements

The authors gratefully acknowledge Steffen Neumann for valuable comments that helped improve the manuscript. J.G. and J.K. are supported by the Research Foundation-Flanders (FWO). A.P. is supported by an Erwin Schrödinger Fellowship of the Austrian Science Fund FWF (J3730-B26). L-C.C. is supported by a fellowship from the Else Kröner-Fresenius-Stiftung. V.L. is supported by a fellowship from the European Research Council (ERC, project No 617393, "CAUSALPATH—Next Generation Causal Analysis project"). The work of R.J.D. is supported by a grant from the National Institutes of Health (R01 CA157996). The work of P.C. is supported by a Belgian Science Policy grant (IUAP P7/03), long-term structural Methusalem funding by the Flemish Government, grants from the FWO and the Foundation Leducq Transatlantic Network (ARTEMIS), FoundationAgainst Cancer, an European Research Council (ERC) Advanced Research Grant (EU-ERC269073), AXA Research Fund and by a Leuven University Fund—Opening the Future.

## Author contributions

JG conceptualized and designed the study, performed most of the data curation, interpreted the data, and wrote the manuscript. AP performed automated searches and did most of the data extraction. L-CC and JK performed data extraction and hand searches of major journals. VL performed statistical analyses, wrote the analyses scripts, and contributed to the manuscript. MD, GE, RJD, and IDW interpreted the data and revised the manuscript. PC directed the study, interpreted the data, and wrote the manuscript.

## Conflict of interest

The authors declare that they have no conflict of interest.

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
