## [Review Process File · EMBO Molecular Medicine]

Meta-analysis of clinical metabolic profiling studies in cancer: challenges and opportunities

Jermaine Goveia, Andreas Pircher, Lena-Christin Conradi, Joanna Kalucka, Vincenzo Lagani, Mieke Dewerchin, Guy Eelen, Ralph J. DeBerardinis, Ian D. Wilson and Peter Carmeliet

Corresponding author: Peter Carmeliet, VIB and KU Leuven

Review timeline:

Submission date:	07 July 2016
Editorial Decision:	01 August 2016
Revision received:	08 August 2016
Accepted:	10 August 2016

Transaction Report:

Editor: Roberto Buccione

1st Editorial Decision

01 August 2016

Thank you again for the submission of your Review article manuscript to EMBO Molecular Medicine. We have now heard back from the three Reviewers whom we asked to evaluate your manuscript.

You will see that all three Reviewers are quite positive and agree that your manuscript is relevant, interesting, useful and well written.

There are a few suggestions for improvement that I am sure you will have no problem dealing with. We would thus be pleased to consider a revised submission, incorporating the reviewers' suggestions. I will be making an editorial decision on your next, final version.

In the likely event of acceptance, you will be asked to fulfill a number of editorial requirements as listed below. I suggest that you provide the following information and amendments requested with the next, final version of your manuscript to accelerate the process:

1) As per our Author Guidelines, the description of all reported data that includes statistical testing must state the name of the statistical test used to generate error bars and P values, the number (n) of independent experiments underlying each data point (not replicate measures of one sample), and the actual P value for each test (not merely 'significant' or 'P < 0.05'). You may provide the P values as a separate table.

2) Every published paper now includes a 'Synopsis' to further enhance discoverability. Synopses are displayed on the journal webpage and are freely accessible to all readers. They include a short

standfirst as well as 2-5 one sentence bullet points that summarise the paper. Please provide the synopsis including the short list of bullet points that summarise the key NEW findings. The bullet points should be designed to be complementary to the abstract - i.e. not repeat the same text. We encourage inclusion of key acronyms and quantitative information. Please use the passive voice. Please attach this information in a separate file or send them by email, we will incorporate it accordingly. You are also welcome to suggest a striking image or visual abstract to illustrate your article. If you do please provide a jpeg file 550 px-wide x 400-px high.

3) Please note that we now mandate that all corresponding authors list an ORCID digital identifier. You may do so through our web platform upon submission and the procedure takes <90 seconds to complete. We also encourage co-authors to supply an ORCID identifier, which will be linked to their name for unambiguous name identification.

I look forward to reading a new revised version of your manuscript as soon as possible.

***** Reviewer's comments *****

Referee #1 (Remarks):

Summary

The current manuscript by Goveia et al. provides a meta-analysis of clinical metabolic profiling (CMP) studies in tumor diseases and diabetes. In this respect, the authors compile CMP studies published between 2010 and 2015. They report that the vast majority of all published CMP studies only report on a subset of measured metabolites and also largely lack appropriate meta-data on patient tumor staging etc. Also, most CMP studies rely on a cross-sectional design, thereby not exploring a longitudinal change in metabolite levels during the course of the disease. In addition, the author's meta-analysis demonstrated that most data remain unconfirmed by independent experimental settings. Due to these limitations, the authors finally employed a semi-quantitative meta-analysis by vote-counting. These analyses demonstrated that across all included CMP studies a number of well-established tumor-associated metabolites, e.g. lactic acid and glutamic acid, could be confirmed to be enriched in tumor tissue. In addition, 3-hydroxybutyric acid could be identified a potential novel tumor marker in cancer patients. Overall, the authors conclude there is a critical need for standardization across future CMP studies.

General Comments

Given the increasing recognition of tumor cell metabolism as a key feature of the malignant phenotype, the identification of tumor-associated metabolites and their potential role as biomarkers and/or therapeutic targets represents an important topic in oncology. In this regard, Goveia et al. provide an interesting and meaningful overview over the validity and usefulness of previously published CMP studies in the field. The manuscript is concise, clearly structured, and comes to clear statements regarding the potential impact of current CMP studies on clinical improvements in tumor diagnostics and therapies. Despite the fact that the chosen approach hardly allows for the discovery of novel metabolite pathways in cancer, the current manuscript may receive broad attention throughout the cancer metabolism community by raising awareness of the weaknesses and limitations of current clinical/experimental approaches. In this respect, the manuscript may serve as an "eye opener" for the cancer metabolite community to increase efforts in data harmonization and reproducibility.

Referee #2 (Remarks):

The manuscript represents an unorthodox and incisive effort to use metabolomic data for meta-analysis. A large portion of the work is a critique of the suitability of the published metabolomic literature for data mining. The authors cogently discuss the limitations of the published literature for this purpose, and make an important comparison to genomic and epigenomic literature. The authors then seek to get around these limitations using a "vote-counting" method. This method allows the authors to identify metabolites that are consistently enriched or depleted in either tumor tissue or

blood from cancer patients, compared to appropriate controls. With this method, the authors identify lactate and glutamate as enriched in tumors and glutamate and 3-hydroxybutyrate as enriched in the blood of cancer patients where tryptophan and glutamine are depleted. While these data are not highly novel, their finding demonstrates the potential of metabolomic meta-analysis, which will be realized more fully when critiques such as this are more widely appreciated.

Issues to be addressed

:

1. The statistical methods are sound, but they are reported only in the Supplement. If these methods could be reported in the main text, it would enhance the paper.
2. The finding of increased lactate is expected and may reasonably be considered to validate the methods. However, the widespread acceptance of the phenomenon of aerobic glycolysis in cancer (which is an acceptance based on strong evidence) may be a source of bias. Studies finding increased lactate may be more likely to be reported or more likely to be published. Studies finding no change in lactate may be less likely to reach the published literature. This source of bias is inherent in meta-analysis and must be discussed.
3. The authors should discuss the possibility that 3-hydroxybutyrate may be elevated in cancer patients due to cachexia.
4. The authors should consider discussing that metabolomic studies are performed with a broader array of technologies than other holistic, non-biased approaches such as transcriptomics. While transcriptomic studies typically rely on either microarray and RNA-seq, metabolomic studies may use a wider variety of analytic methods, complicating direct comparisons between studies.

Referee #3 (Remarks):

This paper by Goveia and coll. entitled "Meta-analysis of clinical metabolic profiling studies in cancer: challenges and opportunities" reports a data mining and semi-quantitative meta-analysis of metabolites comparing healthy and cancer or diabetic patients, to identify distinct metabolite signatures in different pathologies. This study provides the evaluation of the feasibility of this kind of approach and gives some recommendations to improve its clinical impact. This paper is well written and reports important conclusions of clinical importance. Therefore, this work deserves publication in EMM.

Minor issue:

Page 6 lines4-5: the sentence "Surprisingly, ... metabolites" is unclear. Please reformulate.

1st Revision - authors' response

08 August 2016

REFeree #1

Summary

The current manuscript by Goveia et al. provides a meta-analysis of clinical metabolic profiling (CMP) studies in tumor diseases and diabetes. In this respect, the authors compile CMP studies published between 2010 and 2015. They report that the vast majority of all published CMP studies only report on a subset of measured metabolites and also largely lack appropriate meta-data on patient tumor staging etc. Also, most CMP studies rely on a cross-sectional design, thereby not exploring a longitudinal change in metabolite levels during the course of the disease. In addition, the author's meta-analysis demonstrated that most data remain unconfirmed by independent experimental settings. Due to these limitations, the authors finally employed a semi-quantitative meta-analysis by vote-counting. These analyses demonstrated that across all included CMP studies a number of well-established tumor-associated metabolites, e.g. lactic acid and glutamic acid, could

be confirmed to be enriched in tumor tissue. In addition, 3-hydroxybutyric acid could be identified a potential novel tumor marker in cancer patients. Overall, the authors conclude there is a critical need for standardization across future CMP studies.

General Comments

Given the increasing recognition of tumor cell metabolism as a key feature of the malignant phenotype, the identification of tumor-associated metabolites and their potential role as biomarkers and/or therapeutic targets represents an important topic in oncology. In this regard, Gouveia et al. provide an interesting and meaningful overview over the validity and usefulness of previously published CMP studies in the field. The manuscript is concise, clearly structured, and comes to clear statements regarding the potential impact of current CMP studies on clinical improvements in tumor diagnostics and therapies. Despite the fact that the chosen approach hardly allows for the discovery of novel metabolite pathways in cancer, the current manuscript may receive broad attention throughout the cancer metabolism community by raising awareness of the weaknesses and limitations of current clinical/experimental approaches. In this respect, the manuscript may serve as an "eye opener" for the cancer metabolite community to increase efforts in data harmonization and reproducibility.

GENERAL RESPONSE: We thank referee #1 for these thoughtful comments assessing our meta-analysis as a valuable contribution to the field of cancer metabolism.

REFeree #2

The manuscript represents an unorthodox and incisive effort to use metabolomic data for meta-analysis. A large portion of the work is a critique of the suitability of the published metabolomic literature for data mining. The authors cogently discuss the limitations of the published literature for this purpose, and make an important comparison to genomic and epigenomic literature. The authors then seek to get around these limitations using a "vote-counting" method. This method allows the authors to identify metabolites that are consistently enriched or depleted in either tumor tissue or blood from cancer patients, compared to appropriate controls. With this method, the authors identify lactate and glutamate as enriched in tumors and glutamate and 3-hydroxybutyrate as enriched in the blood of cancer patients where tryptophan and glutamine are depleted. While these data are not highly novel, their finding demonstrates the potential of metabolomic meta-analysis, which will be realized more fully when critiques such as this are more widely appreciated.

GENERAL RESPONSE: We thank referee #2 for these generally positive comments. We appreciate the comment to report the statistical methods in the main text, which we originally included in the supplement due to space limitations. We also adapted the discussion as suggested and as detailed below. All changes to the text are marked in red.

Issues to be addressed :

1. The statistical methods are sound, but they are reported only in the Supplement. If these methods could be reported in the main text, it would enhance the paper.

RESPONSE: The statistical methods are now presented in the main text (materials and methods section) (not marked in red).

2. The finding of increased lactate is expected and may reasonably be considered to validate the methods. However, the widespread acceptance of the phenomenon of aerobic glycolysis in cancer (which is an acceptance based on strong evidence) may be a source of bias. Studies finding increased lactate may be more likely to be reported or more likely to be published. Studies finding no change in lactate may be less likely to reach the published literature. This source of bias is inherent in meta-analysis and must be discussed.

RESPONSE: As requested, we now discuss such bias as a potential limitation of our study, but also suggest that this may be partially addressed by full data deposition to public repositories.

3. The authors should discuss the possibility that 3-hydroxybutyrate may be elevated in cancer

patients due to cachexia.

RESPONSE: As requested, the possibility that elevated levels of 3-hydroxybutyrate might be caused by tumor cachexia in affected patients is now discussed in the revised text.

4. The authors should consider discussing that metabolomic studies are performed with a broader array of technologies than other holistic, non-biased approaches such as transcriptomics. While transcriptomic studies typically rely on either microarray and RNA-seq, metabolomic studies may use a wider variety of analytic methods, complicating direct comparisons between studies.

RESPONSE: As requested, we now highlight in the discussion that metabolomics studies are indeed performed with a broad array of technologies, a fact that certainly represents a challenge for inter-study comparisons.

REFeree #3

This paper by Goveia and coll. entitled "Meta-analysis of clinical metabolic profiling studies in cancer: challenges and opportunities" reports a data mining and semi-quantitative meta-analysis of metabolites comparing healthy and cancer or diabetic patients, to identify distinct metabolite signatures in different pathologies. This study provides the evaluation of the feasibility of this kind of approach and gives some recommendations to improve its clinical impact. This paper is well written and reports important conclusions of clinical importance. Therefore, this work deserves publication in EMM.

GENERAL RESPONSE: We appreciate referee #3's comments and assessment of the meta-analysis in terms of clinical importance.

Minor issue:

Page 6 lines4-5: the sentence "Surprisingly, ... metabolites" is unclear. Please reformulate.

RESPONSE: To increase clarity, an additional sentence was included and the sentence referred to was rephrased to read together: "Current metabolic profiling technologies are capable of measuring tens to hundreds of metabolites. However, surprisingly, most individual studies published only a very small subset of all earlier reported metabolites".

Corresponding Author Name: Peter Carmeliet

Manuscript Number: EMM-2016-06798